# Stability analysis and novel solutions to the generalized Degasperis Procesi equation: An application to plasma physics

**S. A. El-Tantawy**[1,2]*, **Alvaro H. Salas**[3], **Castillo H. Jairo E.**[4]

**1** Department of Physics, Faculty of Science, Port Said University, Port Said, Egypt, **2** Research Center for Physics (RCP), Department of Physics, Faculty of Science and Arts, Al-Mikhwah, Al-Baha University, Al-Baha, Saudi Arabia, **3** Department of Mathematics, Universidad Nacional de Colombia, FIZMAKO Research Group, Bogotá, Colombia, **4** Universidad Distrital Francisco José de Caldas, FIZMAKO Research Group, Bogotá, Colombia

* samireltantawy@yahoo.com

**Data Availability Statement:** All data available in the manuscript.

**Funding:** The authors received no specific funding for this study.

## Abstract

In this work two kinds of smooth (compactons or cnoidal waves and solitons) and non-smooth (peakons) solutions to the general Degasperis-Procesi (gDP) equation and its family (Degasperis-Procesi (DP) equation, modified DP equation, Camassa-Holm (CH) equation, modified CH equation, Benjamin-Bona-Mahony (BBM) equation, etc.) are reported in detail using different techniques. The single and periodic peakons are investigated by studying the stability analysis of the gDP equation. The novel compacton solutions to the equations under consideration are derived in the form of Weierstrass elliptic function. Also, the periodicity of these solutions is obtained. The cnoidal wave solutions are obtained in the form of Jacobi elliptic functions. Moreover, both soliton and trigonometric solutions are covered as a special case for the cnoidal wave solutions. Finally, a new form for the peakon solution is derived in details. As an application to this study, the fluid basic equations of a collisionless unmagnetized non-Maxwellian plasma is reduced to the equation under consideration for studying several nonlinear structures in the plasma model.

## 1 Introduction

The study of nonlinear structures is an attractive subject that has captured the minds of many researchers in the twentieth century due to its importance in many fields of science such as optical fiber, Ocean, water tank, physics of plasmas, quantum field theory, Bose-Einstein condensate (BEC), etc. [1–8]. Both ordinary differential equations (ODEs) and partial differential equations (PDEs) have played an effective role in explaining the mechanism and ambiguities of several phenomena that occur in nature on the visible (macroscopic) and invisible (microscopic) levels [9–15]. One of the most important phenomena explained by these equations is the soliton/solitary wave.

The solitary waves are considered one of the most important nonlinear phenomena that have gained their fame over several decades, due to their great importance in transmitting

**Competing interests:** The authors have declared that no competing interests exist.

information to and from transmitting and receiving stations, as well as transmitting information between spacecraft. Solitary waves derive this importance from their properties which they propagate over long distances without losing their energy or changing their shape. Also, they preserve their profile (velocity, shape, energy, etc.) after colliding with each other, and this is considered one of the most important characteristics of solitons. Hence, they have been widely used in optical fibers for many applications. There are several PDEs that can model and simulate solitary waves that propagate in different mediums [16, 17]. To mention but a few, Korteweg-de Vries (KdV) equation is one of the most important and well-known of the equations that were used to describe unmodulated solitons [18, 19]. To this day, and after more than a century has passed since the discovery of the KdV equation, the mathematical and physical analysis behind this equation is the subject of research for many researchers in various fields. The KdV and its family have been used extensively in order to describe various stable structures (solitons, shocks, etc.) that can exist and propagate in several branches of sciences such as optical fibers, seas, oceans, and plasma physics [1–3]. In plasma physics, the KdV equation is devoted to investigate the ion-acoustic (IA) solitons in normal plasma consisting of a collisionless unmagnetized cold ions and warm electrons [20]. Thus many equations have been devoted for modelling solitary waves such as the family of KdV equation, Kadomtsev-Petviashvili (KP) equation, Zakharov-Kuznetsov (ZK) equation, and Benjamin-Bona-Mahony (BBM) equation [1, 2, 21–24]. All these families were used for describing the unmodulated solitons that propagate with phase velocity and with smooth crest. Moreover, the solitary waves that have been described by these families of equations preserve their shapes, velocities, amplitudes, energies after collisions [1, 2, 25–27]. Moreover, these family do not accommodate wave breaking [28]. On the other hand, there is a family of differential equations that is used to describe the modulated solitons (dark solitons, bright solitons, gray solitons) that propagate with the velocity of the group, which is the nonlinear Schrödinger equation and its family [12–16, 29, 30].

There is another class of differential equations that describes the propagation of many nonlinear structures in many different nonlinear and dispersive mediums such as the general Degasperis-Procesi (gDP) equation and its family (Degasperis-Procesi (DP) equation, modified DP equation, Camassa-Holm (CH) equation, modified CH equation, Benjamin-Bona-Mahony (BBM) equation, etc.) [28, 31–35]. The gDP equation and its family are considered good mathematical models for studying the propagation of nonlinear shallow water waves (specially solitary surface wave, peakons, cuspons, periodic waves, and sometimes shock waves) with small amplitude and long wavelength. It has the following general form [31–33]

$$\partial_t(\varphi - \alpha^2\varepsilon^2\partial_x^2\varphi) + \partial_x\big(c_0\varphi + c_1\varphi^2 - c_2\varepsilon^2(\partial_x\varphi)^2 + \varepsilon^2(\gamma - c_3\varphi)\partial_x^2\varphi\big) = 0, \qquad (1)$$

where $\alpha$, $c_0$, $c_1$, $c_2$, $c_3$, and $\gamma$ are real parameters related to the physical problem and $\varepsilon$ is a measure of dispersion. The constants $(\alpha, \gamma) \geq 0$ are associated with different characters of the dispersion semblance. In the Green-Naghdi approximation, the following restriction $\alpha + \gamma = 1/6$ is required. However, Eq (1) has a quite different mathematical properties in the limiting cases $(\alpha, \gamma) = (0, 0)$. In Eq (1) the terms with $(c_2, c_3) \geq 0$ can be treated as representations of nonlinear dispersion. The CH approximation $c_2 + c_3 > 0$ holds [31]. In the text context, we will discuss the family of Eq (1) in detail.

Using different transformations to Eq (1), some types of traveling wave solutions will be investigated and analyzed [36, 37]. In this study, we focused our attention only on the bounded traveling wave solutions which are physical meaningful. Accordingly, some novel solutions such as periodic compactons, cnoidal waves, solitary waves, peakons to Eq (1) will be derived and discussed. It should be mentioned that in the present study, we will obtain some a new

explicit form to the traveling solutions while most published papers only mentioned the conditions for the existence of these solutions and do not provide an explicit picture for these solutions.

## 2 Stability analysis of the gDP equation

For studying the dynamics of Eq (1), let us use the transformation $\varphi \equiv \varphi(\xi)$ where $\xi = (kx + \lambda t + \xi_0)$,

$$\varphi'(c_0 k + \lambda - \epsilon^2(2c_2 + c_3)k^3\varphi'' + 2c_1 k\varphi) + k^2\epsilon^2\varphi'''(\gamma k - \lambda\alpha^2 - c_3 k\varphi) = 0, \tag{2}$$

where primes refer to differentiation with respect to (w.r.t.) $\xi$.

Integrating Eq (2) once over $\xi$, the following ODE with integration constant $D$ is obtained

$$D + (\lambda + kc_0)\varphi + kc_1\varphi^2 - k^3\epsilon^2 c_3\varphi\varphi'' + k^2\epsilon^2(k\gamma - \alpha^2\lambda)\varphi'' - k^3\epsilon^2 c_2(\varphi')^2 = 0, \tag{3}$$

where $D$ is an arbitrary constant of integration.

Eq (3) may be written as

$$\varphi'' = \frac{\epsilon^2 c_2 k^3(\varphi')^2 - \varphi(c_1 k\varphi + c_0 k + \lambda) - D}{k^2\epsilon^2(\gamma k - \lambda\alpha^2 - c_3 k\varphi)}. \tag{4}$$

Eq (4) is a second-order ODE which can be used for studying stability analysis. To do that, we divide Eq (4) into a system of two first-order ODEs as follows:

$$\left.\begin{array}{l} \partial_\xi u \equiv u' = f(u, v) = v, \\[2mm] \partial_\xi v \equiv v' = g(u, v) = \dfrac{\epsilon^2 c_2 k^3 v^2 - u(c_1 ku + c_0 k + \lambda) - D}{k^2\epsilon^2(\gamma k - \lambda\alpha^2 - c_3 ku)}. \end{array}\right\} \tag{5}$$

Here, we used $u \equiv \varphi$ only for beautify, not for something elsewhere. Thereafter, the first derivative for $v$ w.r.t. $u$, reads

$$\partial_u v = \frac{-D - u(\lambda + kc_0 + kuc_1) + k^3\epsilon^2 c_2 v^2}{k^2\epsilon^2(k\gamma - \alpha^2\lambda - kuc_3)v}. \tag{6}$$

The first integration of system (5) with zero constant is obtained by solving the ODE (6). Accordingly, the Hamiltonian is given by

$$H(u, v) = (\mathbb{C}_3 - c_3 ku)^{\frac{2c_2}{c_3}}\left\{v^2 - \frac{1}{c_2\mathbb{C}_1\mathbb{C}_2 k^4\epsilon^2}\left[\mathbb{C}_1\left(\begin{array}{c} 2c_2 k(D + \lambda u) + c_3 Dk \\ +c_0 k(\mathbb{C}_3 + 2c_2 ku) + \lambda\mathbb{C}_3 \end{array}\right) + c_1(c_2 ku(\mathbb{C}_2 ku + 2\mathbb{C}_3) + \mathbb{C}_3^2)\right]\right\}, \tag{7}$$

where $\mathbb{C}_1 = (c_2 + c_3)$, $\mathbb{C}_2 = (2c_2 + c_3)$ and $\mathbb{C}_3 = (\gamma k - \alpha^2\lambda)$.

The Jacobian matrix of system (5) at the equilibrium point $(u, 0) = (u_e, 0)$ is given by

$$A(u_e, 0) = \begin{pmatrix} 0 & -Y \\ 1 & 0 \end{pmatrix}, \tag{8}$$

with $Y = Y_0/Y_1$, where $Y_0 = c_3 k(D - c_1 ku_e^2) + (\gamma k - \alpha^2\lambda)(2c_1 ku_e + c_0 k + \lambda)$ and $Y_1 = k^2\epsilon^2(\alpha^2\lambda + c_3 ku_e - \gamma k)^2$.

It is known that both trace (trace($A$)) and determinant (det($A$)) of the Jacobian matrix (8) are responsible for determining the type of steady state and they read

$$\left.\begin{array}{l} \det(A) = Y, \\ \text{trace}(A) = 0. \end{array}\right\} \tag{9}$$

Note that for $\Delta \equiv (c_0\,k + \lambda)^2 - 4c_1\,Dk \geq 0$, $u_e$ has the following value

$$u_{e1,2} = -\frac{(c_0 k + \lambda) \pm \sqrt{(c_0 k + \lambda)^2 - 4c_1 Dk}}{2c_1 k}, \tag{10}$$

Thus in this case, two equilibrium points for system (5) are obtained as

$$\left.\begin{array}{l} E_{1D} = (u_{e1}, 0), \\ E_{2D} = (u_{e2}, 0). \end{array}\right\} \tag{11}$$

As a particular case, for $D = 0$ and $kc_1(\lambda + kc_0) \neq 0$, the two equilibrium points given in (11) are reduced to

$$\left.\begin{array}{l} E_1 = (0,0), \\ E_2 = \left(-\frac{\lambda + kc_0}{kc_1}, 0\right). \end{array}\right\} \tag{12}$$

It is clear that for $\lambda = -kc_0$, only one equilibrium point exists: $E_1 = (0, 0)$.

The diagnosis of the equilibrium points is based on the sign of det($A$) as follows

$$\left.\begin{array}{l} \det(A) > 0 \Rightarrow \text{a central point,} \\ \det(A) = 0 \Rightarrow \text{uniform motion,} \\ \det(A) < 0 \Rightarrow \text{saddle points.} \end{array}\right\} \tag{13}$$

Observe that sign[det($A$)] = sign[$Y$]. It is known that all traveling wave solutions of system (5) can be determined before solving this system from the form of the phase orbits of system (5) in the phase plane ($u$, $u'$) according to the values of relevant physical parameters: $\alpha$, $c_0$, $c_1$, $c_2$, $c_3$, and $\gamma$. The phase portrait for the above equilibrium points are depicted in Figs 1–3. Fig 1 demonstrates that there are two center equilibrium points for $Y > 0$ and $D = 0$; one of them is at $E_1 = (0, 0)$ and the other at $E_2 = (1.5, 0)$ which det($A$)$_{E_1}$ = 2.14286 and det($A$)$_{E_2}$ = 0.251336 as shown in Fig 1(a). Fig 1(b) and 1(c) represent the profile of the numerical solutions to system (5) at the left and right central points, respectively. For $D \neq 0$ and $Y > 0$, the equilibrium points also becomes a central point at $E_{1D} = (-0.5, 0)$ and $E_{2D} = (2, 0)$ which det($A$)$_{E_{1D}}$ = 0.855373 and det($A$)$_{E_{2D}}$ = 0.30522 as shown in Fig 2. Fig 3 shows that there are two opposite signs for $Y$ which means that one of the equilibrium points is a central ($E_{1D} = (-0.4837, 0)$&det($A$)$_{E_{1D}}$ = 1.95112) and the other is a saddle ($E_{2D} = (1.0337, 0)$&det($A$)$_{E_{2D}}$ = $-23.9443$).

It is clear that system (5) has a singular line at $u = \mathbb{C}_3/(c_3 k)$. Thus, to avoid dividing by zero and the possibility of studying the phase portrait of system (5), the following new transformation is introduced

$$\partial_\xi = k^2 \epsilon^2 (\mathbb{C}_3 - c_3 ku)\partial_\zeta. \tag{14}$$

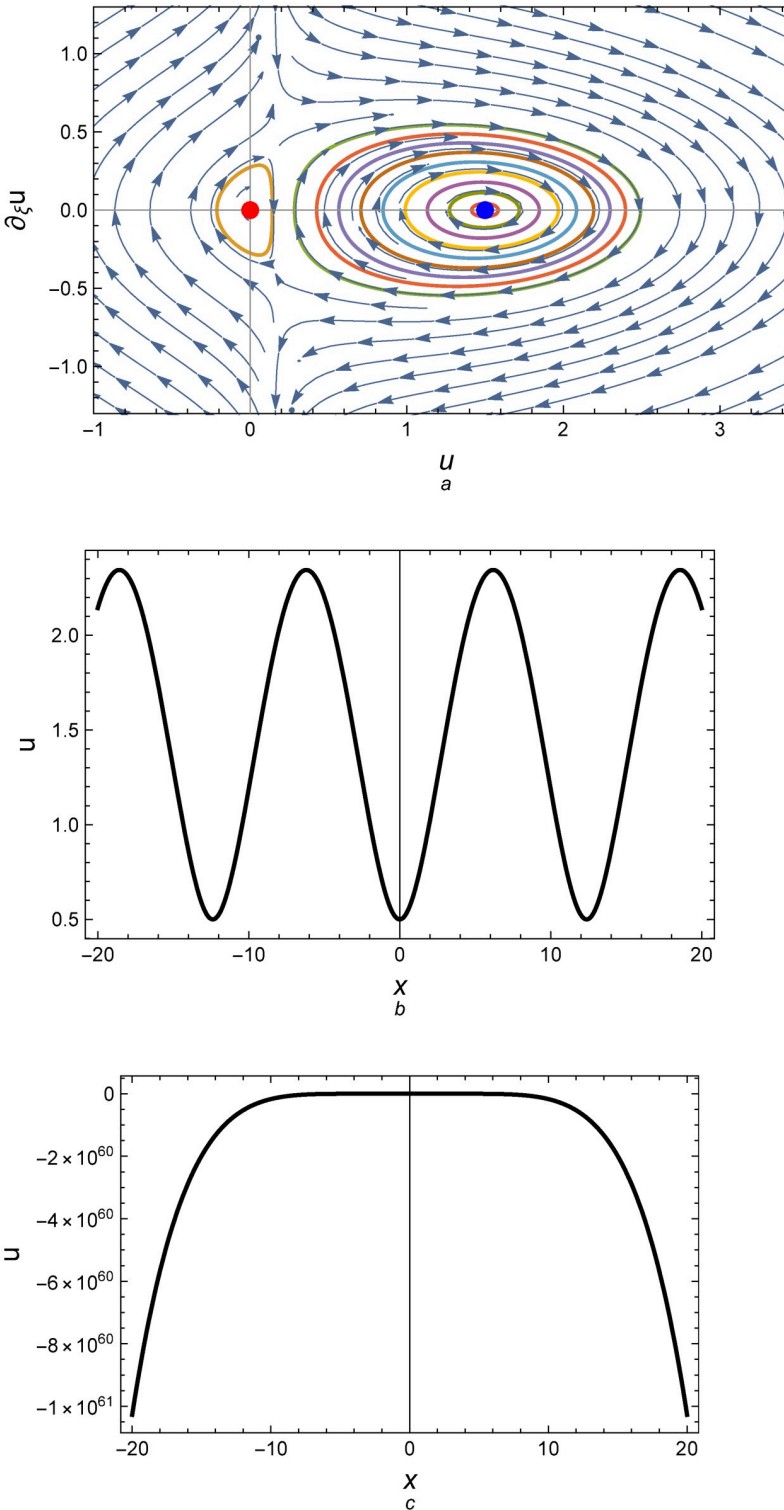

**Fig 1.** (a) Phase portrait for system (5) for the values $c_0 = 1.2$, $c_1 = -1$, $c_2 = 1$, $c_3 = 4.44541$, $D = 0$, $k = 1$, $\alpha = 1$, $\gamma = 1$, $\lambda = 0.3$, $\epsilon = 1$, $A = 0$, and $B = -1$, (b) Profile of the solution according to the central point (Right-Blue point), and (c) Profile of the solution according to the central point (Origin-Red point).

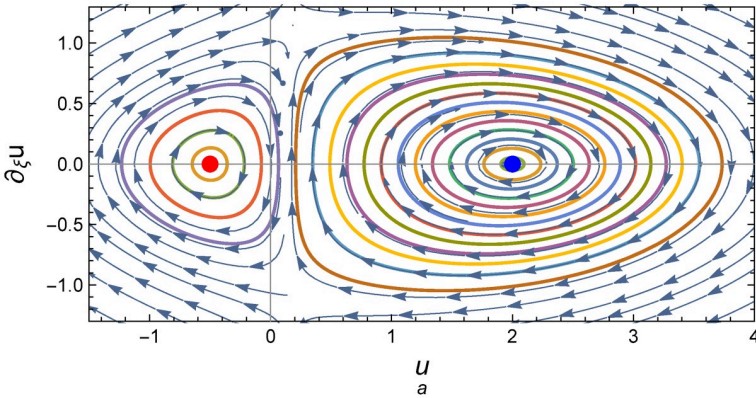

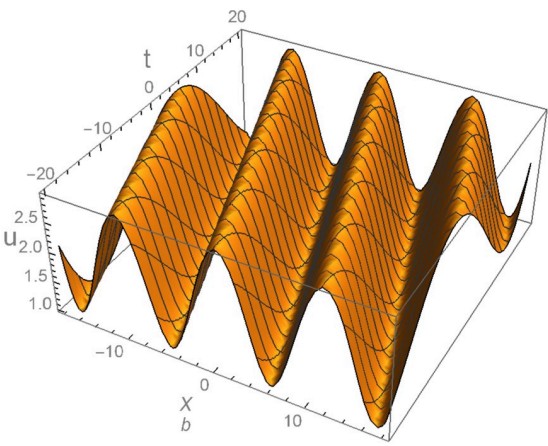

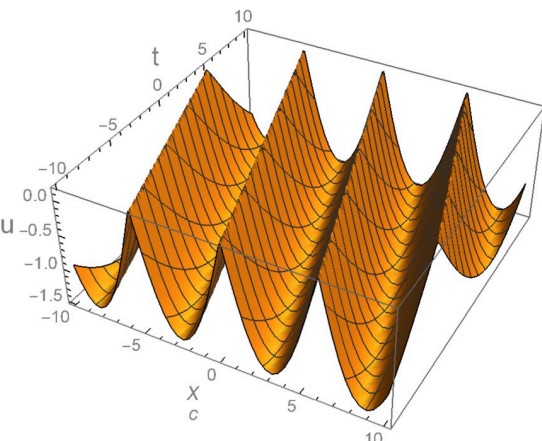

**Fig 2.** (a) Phase portrait for system (5) for the values $c_0 = 1.2$, $c_1 = -1$, $c_2 = 1$, $c_3 = 4.44541$, $D = 1$, $k = 1$, $\alpha = 1$, $\gamma = 1$, $\lambda = 0.3$, $\epsilon = 1$, $A = 0$, and $B = -1$, (b) Profile of the solution according to the central point (Right-Blue point), and (c) Profile of the solution according to the central point (Left-Red point).

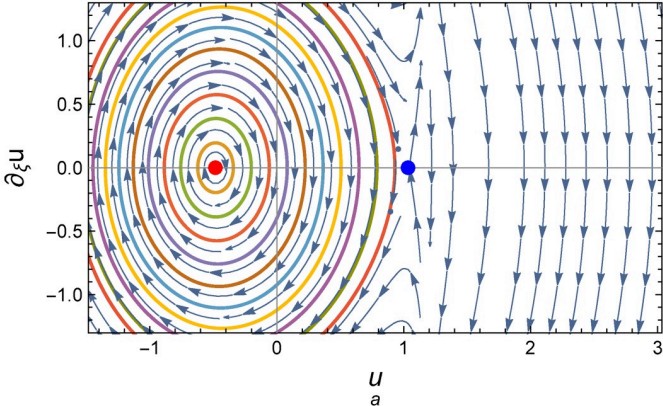

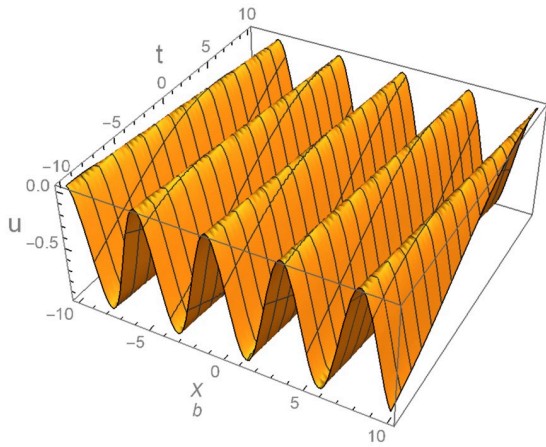

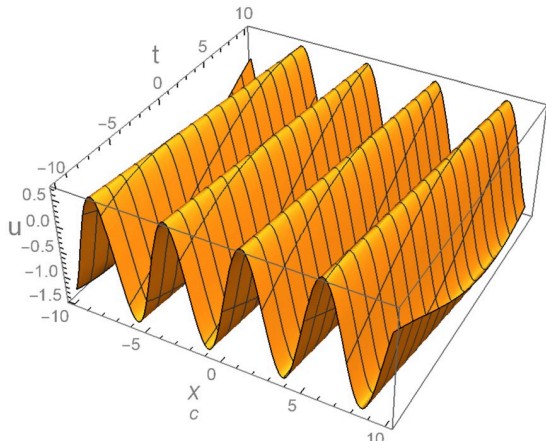

**Fig 3.** (a) Phase portrait for system ([5]) for $c_0 = 1.2$, $c_1 = -2$, $c_2 = 0.1$, $c_3 = 0.941527$, $D = 1$, $k = 1$, $\alpha = 1$, $\gamma = 1$, $\lambda = -0.1$, $\epsilon = 1$, $A = 0$, and $B = -1$, (b) Profile of the solution according to the saddle point (Right-Blue point), and (c) Profile of the solution according to the central point (Left-Red point).

Accordingly, we get

$$\partial_\zeta u = \partial_\xi u \cdot \frac{\partial_\xi}{\partial_\zeta} = F(u, v) = k^2 \epsilon^2 (\mathbb{C}_3 - c_3 ku)v, \tag{15}$$

and

$$\partial_\zeta v = \partial_\xi v \cdot \frac{\partial_\xi}{\partial_\zeta} = G(u, v) = \epsilon^2 c_2 k^3 v^2 - u(c_1 ku + c_0 k + \lambda) - D. \tag{16}$$

Then, system (5) becomes its regular associated system

$$\left.\begin{array}{l} \partial_\zeta u = k^2 \epsilon^2 (\mathbb{C}_3 - c_3 ku)v, \\[4pt] \partial_\zeta v = \epsilon^2 c_2 k^3 v^2 - u(c_1 ku + c_0 k + \lambda) - D. \end{array}\right\} \tag{17}$$

Apparently, the straight line $\gamma k - \lambda \alpha^2 - c_3 ku = 0$, is a solution of system (17). On this straight line, system (17) has two equilibrium points given by

$$(u_e, v_e) = \left( \frac{\mathbb{C}_3}{c_3 k}, \pm \frac{\sqrt{c_3(c_3 Dk + (c_0 k + \lambda)\mathbb{C}_3) + c_1 \mathbb{C}_3^2}}{\sqrt{c_2} c_3 k^2 \epsilon} \right). \tag{18}$$

The dynamics of systems (5) and (17) are different in the neighborhood of the straight line $\gamma k - \lambda \alpha^2 - c_3 ku = 0$. Specially, the variable $\zeta$ is a fast variable while the variable $\xi$ is a slow variable in the sense of the geometric singular perturbation theory [38]. In order to understand the occurrence of "peaked" traveling wave solutions, we must take into consideration both *Theorems A* and *B* that were mentioned in Ref. [38].

The determinant of the Jacobian matrix at the equilibrium points (18) is given by

$$\det_{(u_e, v_e)} = \frac{-2k^2 \epsilon^2}{c_3} \left( \begin{array}{c} k^2 \gamma^2 c_1 - 2k\alpha^2 \gamma \lambda c_1 + \alpha^4 \lambda^2 c_1 + k\gamma\lambda c_3 \\[4pt] -\alpha^2 \lambda^2 c_3 + k^2 \gamma c_0 c_3 - k\alpha^2 \lambda c_0 c_3 + Dkc_3^2 \end{array} \right),$$

and the trace at those points is evaluated as

$$\text{Trace}_{(u_e, v_e)} = \frac{-(c_3 - 2)k\epsilon}{c_3} \sqrt{c_3(c_3 Dk + (c_0 k + \lambda)\mathbb{C}_3) + c_1 \mathbb{C}_3^2}.$$

We also have

$$\lim_{u \to \frac{k\gamma - \alpha^2 \lambda}{c_3 k}} \frac{H(u, v)}{\left( u - \frac{\mathbb{C}_3}{c_3 k} \right)^{2c_2/c_3}}$$

$$= \frac{(c_2 - 1)(\alpha^4 c_1 \lambda^2 - \alpha^2 c_3 \lambda^2 + c_3^2 Dk + \gamma^2 c_1 k^2 + \gamma c_0 c_3 k^2 - 2\alpha^2 \gamma c_1 k\lambda - \alpha^2 c_0 c_3 k\lambda + \gamma c_3 k\lambda)}{2c_2 c_3^2 k^2}.$$

Figs 4 and 5 illustrate the peaked traveling wave solution which correspond to two different parameter sets of values. Fig 4 corresponds to a positive determinant of the Jacobean matrix for system (17), while Fig 5 is obtained for parameters giving a negative determinant.

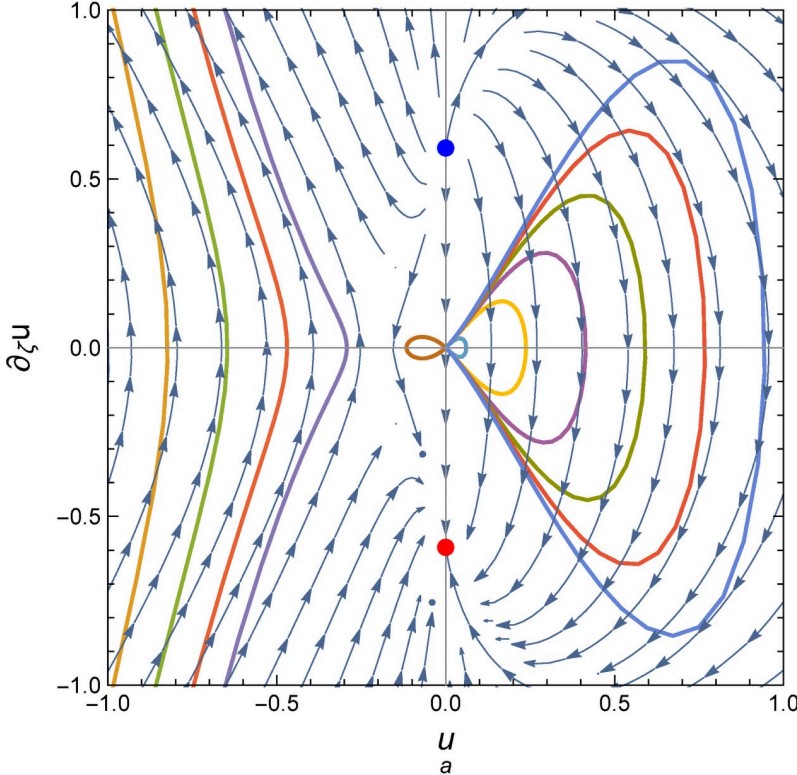

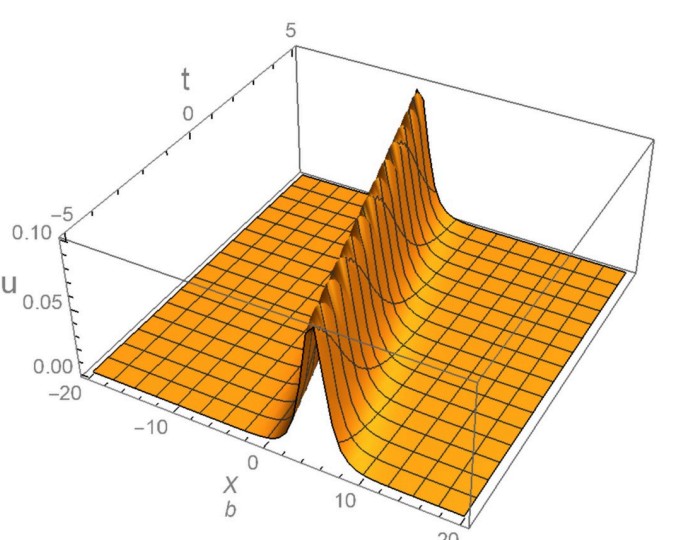

**Fig 4.** Phase portrait for system (17) for the parametric values $c_0 = 1$, $c_1 = -0.2$, $c_2 = 1$, $c_3 = -1.61$, $\alpha = 1$, $\gamma = 1$, $\varepsilon = 1$, $\lambda = 1$, $k = 1$, $D = 0.35$ and (b) Profile of peakon solution for the same parametric values.

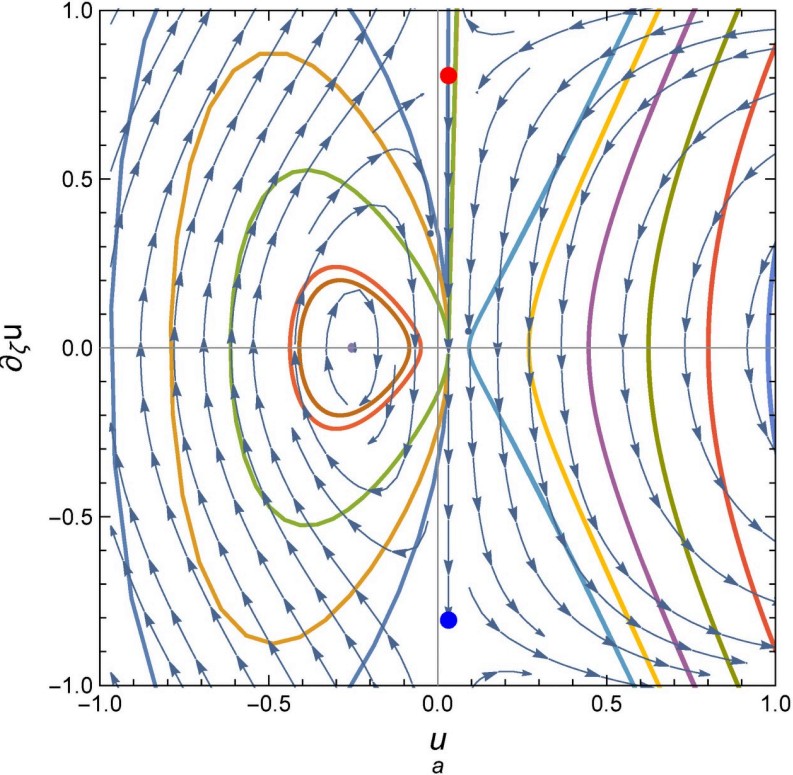

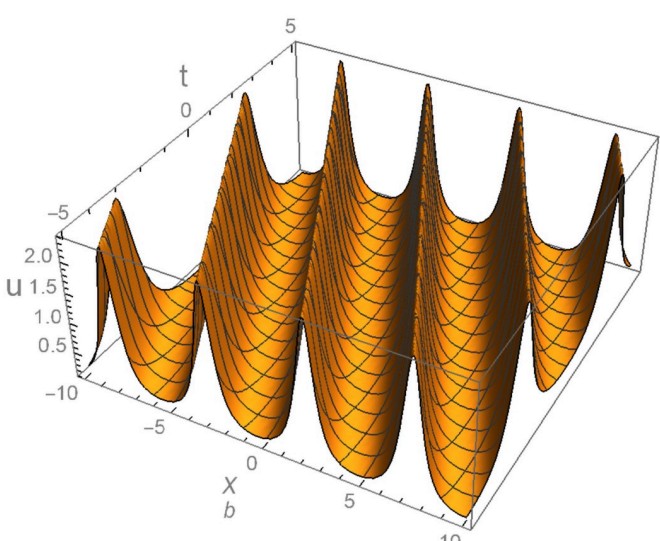

**Fig 5.** Periodic peakon solution for the parametric values $c_0 = 1$, $c_1 = -1.5$, $c_2 = 1$, $c_3 = 1.87316$, $\alpha = 1$, $\gamma = 1$, $\varepsilon = 1$, $\lambda = 0.94$, $k = 1$, $D = 0.59$ and (b) Profile of periodic peakon solution for the same parametric values.

## 3 The methodology for solving the gDP equation

For solving Eq (3), let us rewrite this equation in the form

$$q\psi + p\psi^2 + r\psi \partial_\xi^2 \psi + s \partial_\xi^2 \psi + \delta(\partial_\xi \psi)^2 = D, \tag{19}$$

with $q = (\lambda + kc_0)$, $p = kc_1$, $r = -k^3\, \epsilon^2\, c_3$, $s = k^2\, \epsilon^2(k\gamma - \alpha^2\lambda)$, $\delta = -k^3\, \epsilon^2\, c_2$, and $D$ is an arbitrary constant of integration.

Eq (19) will be solved using two different methods. In the first method, we are going to find a relationship between the gDP equation and the Helmholtz-Duffing (HD) equation which its solutions are well-known. In the second method, a new ansatz will be used for get some solution to the evolution equation in the form of Weiesrtrass elliptic function.

### 3.1 First method: The link between the gDP and HD equations

First, let us find the link between the gDP and HD equations. To do that the traveling wave solution of Eq (19) can be introduced in the following form

$$\psi(\xi) = \frac{1}{w(\xi)}, \tag{20}$$

and by inserting solution (20) into Eq (19), we get

$$\partial_\xi^2 w + F_1 w + F_2 w^2 + F_3 w^3 = 0, \tag{21}$$

with

$$
\begin{aligned}
F_1 &= \frac{p}{\delta + r}, \\
F_2 &= \frac{3(\delta q + qr - ps)}{(\delta + r)(2\delta + r)}, \\
F_3 &= -\frac{2(2\delta^2 D + Dr^2 + 3\delta Dr - ps^2 + qrs + \delta qs)}{\delta(\delta + r)(2\delta + r)},
\end{aligned}
$$

where $w \equiv w(\xi)$ and Eq (21) is called the undamped HD oscillator. It maybe verified that $\psi = \psi(\xi)$ is a solution to (19). We have proved that the solution to Eq (19) is the reciprocal of the solution to some HD Eq (21). The general solution to Eq (21) could be found in details in Refs. [39, 40]. In this case, we can study many nonlinear structures such as solitons, shocks, and cnoidal waves that propagate in plasma physics and optical fiber. For instance, we can reduce the fluid basic equations of some plasma models to Extended Korteweg-de Vries (EKdV) equation (or called Gardner equation) near critical plasma composition and then convert this equation to HD Eq (21) using an appropriate transformation.

**3.1.1 DIAWs in complex plasmas: Connection between HD and EKdV equation.** Let us consider a collisionless unmagnetized plasma composed of heavy particles (here fluid positive ions and immobile dust impurities) and light particles follow superthermal distribution (here, electrons and positrons). It is assumed that the dust ion-acoustic phase velocity is smaller than the thermal velocities of both electron and positron but larger than the ion thermal velocity, where the inertia is provided by the ion mass and the restoring force is provided by the thermal pressures of electrons and positrons. The dynamics of the ion-acoustic structures are governed by the following dimensionless basic equations:

The scaled continuity and momentum equations for positive ions (with labeled "$i$") are, respectively, given by

$$\left.\begin{array}{c} \partial_t n_i + \partial_x(n_i u_i) = 0, \\ \partial_t u_i + u_i \partial_x u_i + \partial_x \phi + 3\sigma_i n_i \partial_x n_i = 0, \end{array}\right\} \tag{22}$$

the scaled number densities of both the superthermal/Kappa electrons (with labeled "$e$") and positrons (with labeled "$p$"), respectively [41, 42]

$$n_e = \mu\left[1 - \frac{\phi}{b_e}\right]^{-\kappa_e + \frac{1}{2}}, \tag{23}$$

and

$$n_p = v\left[1 + \sigma_p \frac{\phi}{b_p}\right]^{-\kappa_p + \frac{1}{2}}, \tag{24}$$

and the Poisson's equation

$$\partial_x^2 \phi = n_e - n_p - n_i + \alpha. \tag{25}$$

Here, $n_i \equiv n_i(x, t)$ and $u_i(x, t)$ represent the normalized number density and fluid velocity of positive ions, respectively, $\phi$ is the normalized electrostatic wave potential, $\sigma_i = T_i/T_e$ gives the ion-to-electron temperature ratio, $\sigma_p = T_e/T_p$ refers to the electron-to-positron temperature ratio, $\mu = n_e^{(0)}/n_i^{(0)}$ represents the electrons concentration, $v = n_p^{(0)}/n_i^{(0)}$ represent the electrons concentration, and $\alpha = Z_d n_d^{(0)}/n_i^{(0)}$ is the negative dust concentration where $n_{i,e,p,d}^{(0)}$ expresses unperturbed/equilibrium number densities to plasma species (ions ("$i$"), electrons ("$e$"), and positrons ("$p$"), and dust particles ("$d$")), $Z_d$ indicates the number of charges that inhabit on the surface of dust impurities, respectively, and the neutrality condition is defined as $\mu + \alpha = 1 + v$. In Kappa distributions, $b_e = (\kappa_e - 3/2)$ and $b_p = (\kappa_p - 3/2)$ where the indices $\kappa_e$ and $\kappa_p$ are a measure of the deviation from thermal (Maxwellian) distribution.

In order to derive Gardner equation, which governs the propagation of small but finite amplitude of dust ion-acoustic waves (DIAWs) in complex plasmas, a reductive perturbation method (RPM) is considered [20]. Accordingly, the following stretching and expansion for the independent and dependent quantities, respectively, are introduced: $[\xi, \tau] = [\epsilon(x - V_{ph} t), \epsilon^3 t]$ and $(n_i, u_i, \phi) = (1, 0, 0) + \sum_{m=1}^{\infty} \epsilon^m(n_i^{(m)}, u_i^{(m)}, \phi^{(m)})$, where $\epsilon$ is a small and real parameter $(0 < \epsilon \ll 1)$ and is a measure of the strength of the dispersion and nonlinearity and $V_{ph}$ donates the normalized phase velocity of the DIAWs. Substituting both the stretching and expansion for the independent and dependent quantities into Eqs (22)–(25) and by following the same procedures in Refs. [43, 44], we finally obtain the EKdV/Gardner equation

$$\partial_\tau \phi + (K_1 \phi + K_2 \phi^2)\partial_\xi \phi + K_3 \partial_\xi^3 \phi = 0, \tag{26}$$

where $\phi \equiv \phi^{(1)}$ and the coefficients of the quadratic nonlinear term $K_1$, cubic nonlinear term

$K_2$, and dispersion term $K_3$ are, respectively, given by

$$K_1 = \frac{1}{2}\left[\frac{(V_{ph}^2 - 3\sigma_i)^2}{V_{ph}}\right],$$

$$K_2 = K_1\left[-2\alpha_2 + \frac{3(V_{ph}^2 + \sigma_i)}{(V_{ph}^2 - 3\sigma_i)^3}\right],$$

$$K_3 = \frac{3}{2}K_1\left[-2\alpha_3 + \frac{5V_{ph}^4 + 30V_{ph}^2\sigma_i + 9\sigma_i^2}{(V_{ph}^2 - 3\sigma_i)^5}\right]$$

with

$$V_{ph} = \sqrt{\frac{1}{\alpha_1} + 3\sigma_i},$$

$$\alpha_1 = \mu\frac{(2\kappa_e - 1)}{(2\kappa_e - 3)} + v\sigma_p\frac{(2\kappa_p - 1)}{(2\kappa_p - 3)},$$

$$\alpha_2 = \mu\frac{(2\kappa_e - 1)(2\kappa_e + 1)}{2(2\kappa_e - 3)^2} - v\sigma_p^2\frac{(2\kappa_p - 1)(2\kappa_p + 1)}{2(2\kappa_p - 3)^2},$$

$$\alpha_3 = \mu\frac{(2\kappa_e - 1)(2\kappa_e + 1)(2\kappa_e + 3)}{6(2\kappa_e - 3)^3} + v\sigma_p^3\frac{(2\kappa_p - 1)(2\kappa_p + 1)(2\kappa_p + 3)}{6(2\kappa_p - 3)^2}.$$

Inserting the transformation $\phi(\xi, \tau) = \phi(\zeta)$ with $\zeta = (\xi + \lambda_f \tau)$ into Eq (25) and integrating the result once over $\zeta$, the following HD equation is obtained [39, 40]

$$\partial_\zeta^2\phi + F_1\phi + F_2\phi^2 + F_3\phi^3 + F_4 = 0, \tag{27}$$

where $\lambda_f$ characterizes the speed of reference frame and it is arbitrary value, $F_1 = \lambda_f / K_3$, $F_2 = K_1 / (2K_3)$, $F_3 = K_2 / (3K_3)$, and $F_4$ represents the constant of integration. Applying the boundary conditions for some nonlinear structures such as solitons and shocks $(\phi, \partial_\zeta\phi, \partial_\zeta^2\phi) \to 0$ as $\zeta| \to \infty$, makes $F_4 = 0$ which leads to the standard HD equation

$$\partial_\zeta^2\phi + F_1\phi + F_2\phi^2 + F_3\phi^3 = 0. \tag{28}$$

By analogy, we find that Eq (27) is the same as Eq (21) where its analytical solutions could be found in details in Ref. [39]. It should be noted that Eq (28) includes a series of nonlinear solutions such as solitons, cnoidal waves, shock waves and all these nonlinear structures are already present in laboratory and space plasma physics.

## 3.2 Second method: The solution of the gDP equation in the form of Weiesrtrass elliptic function

In this section, we look for a solution for Eq (19) in the following ansatz form

$$\psi = A + \frac{B}{1 + C\wp}, \tag{29}$$

where $\wp \equiv \wp(\xi - \xi_0; g_2, g_3)$ represents the Weierstrass elliptic function with invariants $g_2$ and $g_3$. The values of $A$ and $\xi_0$ will be determined later from the initial conditions.

Inserting ansatz (29) into Eq (19) and taking the following relations into account

$$\left.\begin{array}{c} (\partial_\xi \wp)^2 = 4\wp^3 - g_2\wp - g_3, \\[2mm] \partial_\xi^2 \wp = -\dfrac{1}{2}g_2 - \wp^2, \end{array}\right\}$$ (30)

we get

$$\mathbb{R}(t) \equiv \sum_{j=0}^{3} W_j \wp^j = 0,$$ (31)

where $W_j$ is given by

$$
\begin{aligned}
W_0 &= 2A^2p - 4ABC^2g_3r + ABCg_2r + 4ABp + 2Aq - 2B^2C^2\delta g_3 \\
&\quad -4B^2C^2g_3r + B^2Cg_2r + 2B^2p - 4BC^2g_3s + BCg_2s + 2Bq - 2D, \\
W_1 &= -C\left( \begin{array}{c} -8A^2p + 4ABC^2g_3r + 2ABCg_2r - 12ABp - 8Aq + 2B^2C\delta g_2 \\ +3B^2Cg_2r - 4B^2p + 4BC^2g_3s + 2BCg_2s - 6Bq + 8D \end{array} \right), \\
W_2 &= -C\left( \begin{array}{c} -12A^2Cp + 3ABC^2g_2r - 12ABCp + 12ABr - 12ACq \\ -2B^2Cp + 12B^2r + 3BC^2g_2s - 6BCq + 12Bs + 12CD \end{array} \right), \\
W_3 &= -2C^2(-4A^2Cp - 2ABCp + 4ABr - 4ACq - 4B^2\delta - 2B^2r - BCq + 4Bs + 4CD), \\
W_4 &= -2C^3(-A^2Cp - 2ABr - ACq - 2Bs + CD).
\end{aligned}
$$

Equating the coefficients of $\wp^j \equiv \wp^j(\xi; g_2, g_3)$ $(j = 0, 1, 2, 3)$ to zero, we get a system of algebraic equations and by solving this system, the values of the following parameters are obtained

$$
\left\{ \begin{array}{c}
B = -\dfrac{6(Ap(2\delta+r)-ps+q(\delta+r))}{p(2\delta+r)} \,\&\, C = \dfrac{12(\delta+r)}{p}, \\[3mm]
D = \dfrac{q(\delta+r)(s-2A\delta)-p(A^2\delta(2\delta+r)-2A\delta s+s^2)}{(\delta+r)(2\delta+r)}, \\[3mm]
g_2 = \dfrac{p^2}{12(\delta+r)^2} \,\&\, g_3 = \dfrac{p^3}{216(\delta+r)^3},
\end{array} \right.
$$ (32)

where $A$ is an arbitrary constant.

Using the values of the parameters that are defined in Eq (32) in the solution (29), we finally obtain the all traveling wave solutions of the gDP Eq (1) in the following explicit form

$$\varphi_{\text{GDP}} = A + \frac{6[c_1(2Ac_2k + Ac_3k + \gamma k - \alpha^2\lambda) + (c_2 + c_3)(c_0k + \lambda)]}{(2c_2 + c_3)k\left[12(c_2 + c_3)k^2\epsilon^2\wp\left(kx + \lambda t; \frac{c_1^2}{12k^4\epsilon^4(c_2+c_3)^2}, -\frac{c_1^3}{216k^6\epsilon^6(c_2+c_3)^3}\right) - c_1\right]}.$$ (33)

Note that solution (33) is valid only for $c_1(2c_2 + c_3)(c_2 + c_3)k\epsilon \neq 0$. This solution covers several special cases, as we will explain in the next section. It is shown that the solution (33) is periodic and does not decay at infinity. Thus, the periodicity of this solution reads

$$T = \pm 2 \int_{e_1}^{\infty} \frac{1}{\sqrt{4x^3 - g_2x - g_3}}\, dx,$$ (34)

where $g_2$ and $g_3$ are given in Eq (32) and $e_1$ is the greatest root to the following cubic equation

$$4x^3 - g_2x - g_3 = 0.$$ (35)

For $c_1 < 0$ and $c_2 + c_3 > 0$, the periodicity equals

$$T = 2\pi \left| k\varepsilon \sqrt{\frac{c_2 + c_3}{-c_1}} \right|. \tag{36}$$

If $c_1(c_2 + c_3) > 0$, solution (33) becomes unbounded. We conclude that the traveling wave solutions to the gDP Eq (1) are either periodic bounded for $c_1(c_2 + c_3) < 0$ or unbounded for $c_1(c_2 + c_3) > 0$. This means that the gDP Eq (1) does not support soliton solutions unless we impose additional restrictions on its parameters.

### 3.3 Particular cases: Traveling wave solutions of the family of the gDP equation

As we mentioned above that the gDP Eq (1) is a general PDE and under certain assumptions, it can be reduced to many known PDEs such as KdV-type equation, BBM equation, CH equation, and Degasperis-Procesi (DP) equation. First, it is assumed that the solution of the equations under consideration is given by Eq (29) and by substituting this relation into the equation under consideration, and by following the same procedure that was used in solving Eq (1), we finally obtain the values of $g_2$, $g_3$, $C$, and sometimes $B$ for all cases as follow:

(i). For $\alpha = c_2 = c_3 = 0$, Eq (1) reduces to the following KdV-type equation

$$\partial_t \varphi + c_0 \partial_x \varphi + 2c_1 \varphi \partial_x \varphi + \gamma \varepsilon^2 \partial_x^3 \varphi = 0. \tag{37}$$

The coefficients of its periodic solution read

$$\begin{cases} C = \frac{12k^3 \gamma \epsilon^2}{\lambda + kc_0 + 2Akc_1}, \\ g_2 = \frac{(\lambda + kc_0 + 2Akc_1)(3\lambda + 3kc_0 + 2(3A+B)kc_1)}{36k^6 \gamma^2 \epsilon^4}, \\ g_3 = \frac{(\lambda + kc_0 + 2Akc_1)^2(\lambda + kc_0 + (2A+B)kc_1)}{216k^9 \gamma^3 \epsilon^6}. \end{cases} \tag{38}$$

(ii). For $\gamma = c_2 = c_3 = 0$, Eq (1) is reduced to the well known BBM equation

$$\partial_t(\varphi - \alpha^2 \varepsilon^2 \partial_x^2 \varphi) + \partial_x(c_0 \varphi + c_1 \varphi^2) = 0. \tag{39}$$

The coefficients of its solution read

$$\begin{cases} C = -\frac{12\alpha^2 k^2 \lambda \epsilon^2}{2Ac_1 k + c_0 k + \lambda}, \\ g_2 = \frac{(\lambda + kc_0 + 2Akc_1)(3\lambda + 3kc_0 + 2(3A+B)kc_1)}{36k^4 \alpha^4 \epsilon^4 \lambda^2}, \\ g_3 = -\frac{(\lambda + kc_0 + 2Akc_1)^2(\lambda + kc_0 + (2A+B)kc_1)}{216k^6 \alpha^6 \epsilon^6 \lambda^3}. \end{cases} \tag{40}$$

(iii). For $c_1 = 3c_3/2\alpha^2$, $c_2 = c_3/2$, and $\gamma = 0$, Eq (1) reduces to the CH equation as

$$\partial_t\left(\varphi - \alpha^2 \varepsilon^2 \partial_x^2 \varphi\right) + \partial_x\left(c_0 \varphi + \frac{3c_3}{2\alpha^2} \varphi^2 - \frac{c_3}{2} \varepsilon^2 (\partial_x \varphi)^2 + \varepsilon^2(-c_3 \varphi)\partial_x^2 \varphi\right) = 0. \tag{41}$$

The coefficients of its solution read

$$
\begin{cases}
B = -\frac{3(2Ac_3 + \alpha^2 c_0)}{c_3}, \\[2mm]
C = -12\alpha^2 k^2 \varepsilon^2, \\[2mm]
g_2 = \frac{1}{12k^4 \alpha^4 \varepsilon^4}, \\[2mm]
g_3 = -\frac{1}{216 k^6 \alpha^6 \varepsilon^6}.
\end{cases}
\tag{42}
$$

(iv). In the case $c_2 = c_3$, $c_1 = 2c_3/\alpha^2$, and $c_0 = \gamma = 0$, Eq (1) reduces to the DP equation [28, 33, 45]:

$$
\partial_t \left( \varphi - \alpha^2 \varepsilon^2 \partial_x^2 \varphi \right) + \partial_x \left( \frac{2c_3}{\alpha^2} \varphi^2 - c_3 \varepsilon^2 (\partial_x \varphi)^2 - c_3 \varepsilon^2 \varphi \partial_x^2 \varphi \right) = 0.
\tag{43}
$$

The coefficients of its solution read

$$
\begin{cases}
B = -6A, \\[2mm]
C = -12\alpha^2 k^2 \epsilon^2, \\[2mm]
g_2 = \frac{1}{12k^4 \alpha^4 \epsilon^4}, \\[2mm]
g_3 = -\frac{1}{216 k^6 \alpha^6 \epsilon^6}.
\end{cases}
\tag{44}
$$

## 4 Cnoidal, soliton, peakon, and trigonometric solutions

In this section, we will derive and discuss two types of cnoidal wave solution and soliton solution to the family of gDP Eq (1) under two different conditions: (i) $c_2 = -2c_3$ and (ii) $c_3 = -2c_2/3$.

(i) First case: when $c_2 = -2c_3$, the following ansatz is introduced

$$
\varphi = A + B\mathrm{cn}[\beta(x - Vt + X_0), m],
\tag{45}
$$

and by inserting this ansatz into Eq (1), we get

$$
\begin{aligned}
&Bm\beta^2\varepsilon^2(c_2 + 2c_3)\mathrm{cn}^3 - 6m\beta^2\varepsilon^2(V\alpha^2 + \gamma - Ac_3)\mathrm{cn}^2 + \\
&B(2c_1 - m\beta^2\varepsilon^2 c_2 - 2\beta^2\varepsilon^2 c_3 + 2m\beta^2\varepsilon^2 c_3)\mathrm{cn} + \\
&A\beta^2 c_3(1 - 2m)\varepsilon^2 + 2Ac_1 + c_0 + \\
&\beta^2(2m - 1)\varepsilon^2(\gamma + \alpha^2 V) - V = 0,
\end{aligned}
\tag{46}
$$

where $\varphi \equiv \varphi(x, t)$ and $\mathrm{cn} \equiv \mathrm{cn}[\beta(x - Vt + \xi_0), m]$.

Equating all coefficients of $\mathrm{cn}^j$ ($j = 0, 1, 2, 3$) to zero, an algebraic system of equations is obtained and by solving it, we get

$$
\begin{cases}
\beta = \frac{\sqrt{c_1}}{\sqrt{c_2 m\epsilon^2 + c_3 \varepsilon^2}}, \\[2mm]
V = \frac{2\gamma c_1 + c_0 c_3}{c_3 - 2\alpha^2 c_1}, \\[2mm]
A = \frac{\alpha^2 c_0 + \gamma}{c_3 - 2\alpha^2 c_1}.
\end{cases}
\tag{47}
$$

Thus, the cnoidal wave solution to Eq (1) reads

$$\varphi_{Cn} = \frac{\alpha^2 c_0 + \gamma}{c_3 - 2\alpha^2 c_1} + B\mathrm{cn}\left[\frac{1}{\varepsilon}\sqrt{\frac{c_1}{c_2 m + c_3}}\left(x - \frac{2\gamma c_1 + c_0 c_3}{c_3 - 2\alpha^2 c_1}t + X_0\right), m\right],$$ (48)

with $B \neq 0$.

As a special case, solution (48) reduces to the soliton solution when $m \to 1$

$$\varphi_{Sol} = \frac{\alpha^2 c_0 + \gamma}{c_3 - 2\alpha^2 c_1} + B\mathrm{sech}\left[\frac{1}{\varepsilon}\sqrt{\frac{c_1}{c_2 + c_3}}\left(x - \frac{2\gamma c_1 + c_0 c_3}{c_3 - 2\alpha^2 c_1}t + X_0\right)\right].$$ (49)

Furthermore, when $m \to 0$, the trigonometric solution is covered

$$\varphi_{Trigo} = \frac{\gamma + \alpha^2 c_0}{c_3 - 2\alpha^2 c_1} + B\cos\left[\frac{1}{\varepsilon}\sqrt{\frac{c_1}{c_3}}\left(x + X_0 - \frac{(2\gamma c_1 + c_0 c_3)t}{-2\alpha^2 c_1 + c_3}\right)\right].$$ (50)

(ii) Second case: when $c_3 = -2c_2/3$, the following ansatz is introduced

$$\varphi(x, t) = A + B\mathrm{cn}^2[\beta(x - Vt + X_0), m].$$ (51)

Inserting this ansatz into Eq (1) gives us

$$3V + 12V\alpha^2\beta^2\varepsilon^2 - 24mV\alpha^2\beta^2\varepsilon^2 + 12\beta^2\gamma\epsilon^2 - 24m\beta^2\gamma\epsilon^2 - 3c_0$$
$$-6Ac_1 + 8A\beta^2\varepsilon^2 c_2 + 8B\beta^2\varepsilon^2 c_2 - 16Am\beta^2\varepsilon^2 c_2 - 8Bm\beta^2\varepsilon^2 c_2 -$$
$$2\left(\begin{array}{c}-18mV\alpha^2\beta^2\varepsilon^2 - 18m\beta^2\gamma\epsilon^2 + 3Bc_1 + 4B\beta^2\varepsilon^2 c_2 - \\ 12Am\beta^2\varepsilon^2 c_2 - 8Bm\beta^2\varepsilon^2 c_2\end{array}\right)\mathrm{cn}^2 = 0.$$ (52)

Equating the coefficients of $\mathrm{cn}^j$ $(j = 0, 2)$ to zero gives us an algebraic system of equations. The solution of this system gives

$$V = \frac{2\beta^2 c_2 \varepsilon^2 (9c_0 m - 8\beta^2 Bc_2((m-1)m+1)\varepsilon^2) + 9Bc_1^2 - 54\beta^2\gamma c_1 m\varepsilon^2}{18\beta^2 m\varepsilon^2(3\alpha^2 c_1 + c_2)},$$ (53)

and

$$A = \left[\frac{1}{12\beta^2 m\varepsilon^2(3\alpha^2 c_1 + c_2)}\right]\left[\begin{array}{c}4\beta^2 Bc_2\varepsilon^2(4\alpha^2\beta^2((m-1)m+1)\varepsilon^2 - 2m + 1) + \\ 3Bc_1(4\alpha^2\beta^2(1-2m)\varepsilon^2 + 1) - 18\beta^2 m\varepsilon^2(\alpha^2 c_0 + \gamma)\end{array}\right].$$ (54)

As a special case, when $m \to 1$, the soliton solution is covered

$$\varphi_{Sol} = \frac{B(4\alpha^2\beta^2\varepsilon^2 - 1)(4\beta^2 c_2\varepsilon^2 - 3c_1) - 18\alpha^2\beta^2 c_0\varepsilon^2 - 18\beta^2\gamma\varepsilon^2}{12\beta^2\varepsilon^2(3\alpha^2 c_1 + c_2)} +$$
$$B\mathrm{sech}^2\left(\beta\left(x + \frac{(54\beta^2\gamma\varepsilon^2 c_1 - 9Bc_1^2 + 2\beta^2\varepsilon^2 c_2(-9c_0 + 8B\beta^2\varepsilon^2 c_2))t}{18\beta^2\varepsilon^2(3\alpha^2 c_1 + c_2)} + \xi_0\right)\right).$$ (55)

For $m \to 0$, the solution degenerates to a constant.

In order to find a peakon solution of Eq (1), the following ansatz is considered

$$\varphi = A\exp[\beta(x - Vt + \eta_0)],$$ (56)

with $A\beta V \neq 0$.

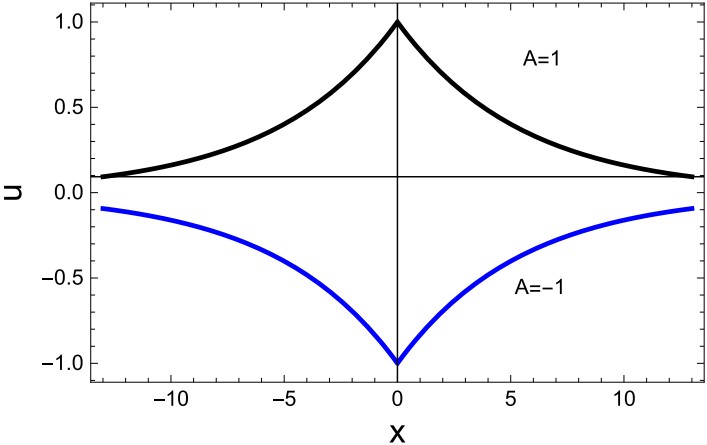

**Fig 6. Profile of peakon solution (59) is plotted in the plane $(x, t)$ for $\alpha = 1$, $\gamma = 0$, $\varepsilon = 1$, $c_0 = c_2 = 1$, $c_1 = 3$, $c_3 = 5c_2$, and $\eta_0 = 0$.**

Using this ansatz in Eq (1), gives

$$A\beta(-\alpha^2\beta^2\lambda\epsilon^2 + \beta^2\gamma\epsilon^2 + c_0 + \lambda)z - 2A^2\beta(\beta^2 c_2\epsilon^2 + \beta^2 c_3\epsilon^2 - c_1)z^2 = 0 \tag{57}$$

Equating the coefficients of $z = \exp\left[\beta(x - Vt + \eta_0)\right]$ and $z^2 = \exp\left[2\beta(x - Vt + \eta_0)\right]$ to zero gives a system of algebraic equations. By solving this system, we obtain

$$\begin{cases} \lambda = \frac{\gamma c_1 + c_0(c_2 + c_3)}{c_2 + c_3 - \alpha^2 c_1}, \\ \beta = \pm\frac{1}{\epsilon}\sqrt{\frac{c_1}{c_2 + c_3}}. \end{cases} \tag{58}$$

Then the peakon solutions are given by

$$\varphi_{\text{Peak}} = A\exp\left[-\left|\frac{1}{\epsilon}\sqrt{\frac{c_1}{c_2 + c_3}}\left(x - \frac{\gamma c_1 + c_0(c_2 + c_3)}{c_2 + c_3 - \alpha^2 c_1}t + \eta_0\right)\right|\right], \tag{59}$$

for $A \neq 0$. Fig 6 shows the profile of peakon solution (59).

## 5 Conclusion

In this paper, some new analytical solutions in terms of the Weierstrass elliptic double periodic function and Jacobi elliptic function to the generalized Degasperis Procesi (gDP) equation and its family are constructed in an explicit form. Moreover, the stability analysis of gDP equation has been investigated. According to this method, the single and periodic peakons are investigated. Afterthought, three techniques are devoted to derive and obtain an explicit formula for the traveling wave solutions to gDP equation. In the first method, the gDP equation was reduced to the undamped Helmholtz-Duffing (HD) oscillator using an appropriate transformation whose solutions were known in the literature. According to this method, we can investigate many nonlinear structures in different models of plasma physics by reducing the fluid equations of plasma particles to the Gardner equation and after that converting Gardner equation to HD equation. As for the second technique, a new ansatz is introduced to find novel solutions to gDP equation and its family in the form of Weierstrass elliptic function. Also, the periodicity of this solution is obtained. In the third technique, the cnoidal, soliton, and trigonometric solutions for the gDP equation are investigated in the form of Jacobi elliptic

functions. Finally, an explicit form for peakon solution is derived in details for the equation under consideration. All obtained solutions may be important in explaining many mysterious phenomena that have no interpretation and appear in nonlinear media such as Ocean, plasma physics, and optical fibers.

## Author Contributions

**Conceptualization:** S. A. El-Tantawy.

**Data curation:** S. A. El-Tantawy.

**Formal analysis:** S. A. El-Tantawy, Castillo H. Jairo E.

**Investigation:** S. A. El-Tantawy, Castillo H. Jairo E.

**Methodology:** S. A. El-Tantawy, Alvaro H. Salas, Castillo H. Jairo E.

**Resources:** S. A. El-Tantawy.

**Software:** S. A. El-Tantawy.

**Supervision:** Alvaro H. Salas.

**Writing – original draft:** S. A. El-Tantawy, Castillo H. Jairo E.

**Writing – review & editing:** S. A. El-Tantawy.

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
