## [Decision Letter · Decision Letter 0]

28 May 2021

PONE-D-21-15906

Stability analysis and novel solutions to the generalizaed Degasperis Procesi equation: An application to plasma physics

PLOS ONE

Dear Dr. Samir A. El-Tantawy,

Thank you for submitting your manuscript to PLOS ONE. After careful consideration, we feel that it has merit but does not fully meet PLOS ONE’s publication criteria as it currently stands. Therefore, we invite you to submit a revised version of the manuscript that addresses the points raised during the review process.

Follow the suggestions of the two reviewers and carefully revise the manuscript one by one.

We look forward to receiving your revised manuscript.

Kind regards,

Shou-Fu Tian

Academic Editor

PLOS ONE

Journal Requirements:

"Taif University Researchers Supporting Project number (TURSP-2020/275), Taif Uni-

versity, Taif, Saudi Arabia."

"All authors contributed equally and significantly in writing this article. All authors read and approved the final manuscript. "

3. Please ensure that you refer to Figure 6 in your text as, if accepted, production will need this reference to link the reader to the figure.

Additional Editor Comments:

Follow the suggestions of the two reviewers and carefully revise the manuscript one by one.

Reviewer 1： In this study, the authors have studied two kinds of smooth (compactons or cnoidal waves and solitons) and nonsmooth (peakons) solutions to the general Degasperis-Procesi (gDP) equation and its family using different techniques. Moreover, the stability analysis of gDP equation is investigated. According to this method, the single and periodic peakons are investigated. Afterthought, three techniques are devoted to derive and obtain an explicit formula for the traveling wave solutions to gDP equation. The results are corrects, and important for the understanding of such equations.

This is a very well written manuscript as I read and understood. The contents of this paper are interesting and helpful for wide audience. If the authors discuss the following latest progress of soliton solution in the section of INTRODUCTION, this paper will be more excellent.

[1] Stability analysis, solitary wave and explicit power series solutions of a (2 + 1)-dimensional nonlinear Schrödinger equation in a multicomponent plasma, International Journal of Numerical Methods for Heat & Fluid Flow, 3(5), (2021) 1732-1748.

[2] Stability analysis solutions, optical solitons, Gaussian solutions and traveling wave solutions of the nonlinear Schrdinger governing equation. Optik - International Journal for Light and Electron Optics, 2017, 158:391-398.

[3] Riemann‐Hilbert approach for multisoliton solutions of generalized coupled fourth‐order nonlinear Schrödinger equations, Math. Meth. Appl. Sci. 43 (2) (2020) 865-880

[4]Dynamics of the soliton waves, breather waves, and rogue waves to the cylindrical Kadomtsev-Petviashvili equation in pair-ion–electron plasma, Phys. Fluids 31 (10) (2019) 102107

[5] Optical soliton solutions, periodic wave solutions and complexitons of the cubic Schrdinger equation with a bounded potential. Superlattices and Microstructures, 2018, 113:510-518.

[6] Integrability, soliton solutions and modulation instability analysis of a (2+ 1)-dimensional nonlinear Heisenberg ferromagnetic spin chain equation, Comput. Math. Appl. 77 (3) (2019) 770-778.

Authors should carefully discuss these references in their introduction. I highly recommend the manuscript to be published after the minor modification.

Reviewer 2：The manuscript is devoted to a theoretical study of the generalizaed Degasperis-Procesi equation. The authors have studied a direct way, which is used to explicitly construct some exact solutions of the equation. Finally, the main characteristics of these solutions are discussed with some graphics. These results may contribute to enriching the corresponding nonlinear wave phenomena. The topic is of current interest in soliton theory and such a study is a good supplement to the existing literature on exact solutions to nonlinear equations. The authors should take a look of Stud. Appl. Math. 132 (2014) 212-246 ; J. Phys. A: Math. Theor. 45 (2012) 055203; Proc. Am. Math. Soc. 146 (2018) 3353-3365; J. Math. Phys. 59 (2018) 073505; J. Phys. Soc. Jpn. 89（2020）124003; Europhys. Lett. 126 (2019) 15001. It should be helpful for you to understand the nonlinear wave equations with their exact solutions.

It is expected that the authors could revise and/or amend their manuscript to comment or remark on those existing references on related studies.

In my opinion, their results provided in this manuscript can be used to enrich the dynamical behavior of nonlinear wave fields. In conclusion, I recommend a thorough revision of this manuscript before deciding on a publication.

Reviewers' comments:

Reviewer's Responses to Questions

**Comments to the Author**

1. Is the manuscript technically sound, and do the data support the conclusions?

Reviewer #1: Yes

Reviewer #2: Yes

2. Has the statistical analysis been performed appropriately and rigorously? 

Reviewer #1: Yes

Reviewer #2: Yes

3. Have the authors made all data underlying the findings in their manuscript fully available?

Reviewer #1: Yes

Reviewer #2: Yes

4. Is the manuscript presented in an intelligible fashion and written in standard English?

Reviewer #1: Yes

Reviewer #2: Yes

5. Review Comments to the Author

Reviewer #1: In this study, the authors have studied two kinds of smooth (compactons or cnoidal waves and solitons) and nonsmooth (peakons) solutions to the general Degasperis-Procesi (gDP) equation and its family using different techniques. Moreover, the stability analysis of gDP equation is investigated. According to this method, the single and periodic peakons are investigated. Afterthought, three techniques are devoted to derive and obtain an explicit formula for the traveling wave solutions to gDP equation. The results are corrects, and important for the understanding of such equations.

This is a very well written manuscript as I read and understood. The contents of this paper are interesting and helpful for wide audience. If the authors discuss the following latest progress of soliton solution in the section of INTRODUCTION, this paper will be more excellent.

[1] Stability analysis, solitary wave and explicit power series solutions of a (2 + 1)-dimensional nonlinear Schrödinger equation in a multicomponent plasma, International Journal of Numerical Methods for Heat & Fluid Flow, 3(5), (2021) 1732-1748.

[2] Stability analysis solutions, optical solitons, Gaussian solutions and traveling wave solutions of the nonlinear Schrdinger governing equation. Optik - International Journal for Light and Electron Optics, 2017, 158:391-398.

[3] Riemann‐Hilbert approach for multisoliton solutions of generalized coupled fourth‐order nonlinear Schrödinger equations, Math. Meth. Appl. Sci. 43 (2) (2020) 865-880

[4]Dynamics of the soliton waves, breather waves, and rogue waves to the cylindrical Kadomtsev-Petviashvili equation in pair-ion–electron plasma, Phys. Fluids 31 (10) (2019) 102107

[5] Optical soliton solutions, periodic wave solutions and complexitons of the cubic Schrdinger equation with a bounded potential. Superlattices and Microstructures, 2018, 113:510-518.

[6] Integrability, soliton solutions and modulation instability analysis of a (2+ 1)-dimensional nonlinear Heisenberg ferromagnetic spin chain equation, Comput. Math. Appl. 77 (3) (2019) 770-778.

Authors should carefully discuss these references in their introduction. I highly recommend the manuscript to be published after the minor modification.

Reviewer #2: The manuscript is devoted to a theoretical study of the generalizaed Degasperis-Procesi equation. The authors have studied a direct way, which is used to explicitly construct some exact solutions of the equation. Finally, the main characteristics of these solutions are discussed with some graphics. These results may contribute to enriching the corresponding nonlinear wave phenomena. The topic is of current interest in soliton theory and such a study is a good supplement to the existing literature on exact solutions to nonlinear equations. The authors should take a look of Stud. Appl. Math. 132 (2014) 212-246 ; J. Phys. A: Math. Theor. 45 (2012) 055203; Proc. Am. Math. Soc. 146 (2018) 3353-3365; J. Math. Phys. 59 (2018) 073505; J. Phys. Soc. Jpn. 89（2020）124003; Europhys. Lett. 126 (2019) 15001. It should be helpful for you to understand the nonlinear wave equations with their exact solutions.

It is expected that the authors could revise and/or amend their manuscript to comment or remark on those existing references on related studies.

In my opinion, their results provided in this manuscript can be used to enrich the dynamical behavior of nonlinear wave fields. In conclusion, I recommend a thorough revision of this manuscript before deciding on a publication.

6. PLOS authors have the option to publish the peer review history of their article (what does this mean?). If published, this will include your full peer review and any attached files.

Reviewer #1: No

Reviewer #2: No

---

## [Decision Letter · Decision Letter 1]

5 Jul 2021

Stability analysis and novel solutions to the generalized Degasperis Procesi equation: An application to plasma physics

PONE-D-21-15906R1

Dear Dr.  El-Tantawy ,

We’re pleased to inform you that your manuscript has been judged scientifically suitable for publication and will be formally accepted for publication once it meets all outstanding technical requirements.

Kind regards,

Shou-Fu Tian

Academic Editor

PLOS ONE

Additional Editor Comments (optional):

Accept

Reviewers' comments:

Reviewer's Responses to Questions

**Comments to the Author**

1. If the authors have adequately addressed your comments raised in a previous round of review and you feel that this manuscript is now acceptable for publication, you may indicate that here to bypass the “Comments to the Author” section, enter your conflict of interest statement in the “Confidential to Editor” section, and submit your "Accept" recommendation.

Reviewer #1: All comments have been addressed

Reviewer #2: All comments have been addressed

2. Is the manuscript technically sound, and do the data support the conclusions?

Reviewer #1: Yes

Reviewer #2: Yes

3. Has the statistical analysis been performed appropriately and rigorously? 

Reviewer #1: Yes

Reviewer #2: Yes

4. Have the authors made all data underlying the findings in their manuscript fully available?

Reviewer #1: Yes

Reviewer #2: Yes

5. Is the manuscript presented in an intelligible fashion and written in standard English?

Reviewer #1: Yes

Reviewer #2: Yes

6. Review Comments to the Author

Reviewer #1: Authors have improved the manuscript and also the current research work contains fruitful results. Thus I strongly recommend its publication.

Reviewer #2: Review Comments to the Author:

The revised manuscript is very good, I think that the work should be published in plos one.

7. PLOS authors have the option to publish the peer review history of their article (what does this mean?). If published, this will include your full peer review and any attached files.

Reviewer #1: No

Reviewer #2: No

---

## [Editor Report · Acceptance letter]

3 Sep 2021

PONE-D-21-15906R1 

Stability analysis and novel solutions to the generalized Degasperis Procesi equation: An application to plasma physics 

Dear Dr. El-Tantawy:

I'm pleased to inform you that your manuscript has been deemed suitable for publication in PLOS ONE. Congratulations! Your manuscript is now with our production department. 

Kind regards, 

on behalf of

Dr. Shou-Fu Tian 

Academic Editor

PLOS ONE